# Iridium-Based Nanohybrids: Synthesis, Characterization, Optical Limiting, and Nonlinear Optical Properties

**DOI:** 10.3390/nano13142131

**Published:** 2023-07-22

**Authors:** Nikolaos Chazapis, Michalis Stavrou, Georgia Papaparaskeva, Alexander Bunge, Rodica Turcu, Theodora Krasia-Christoforou, Stelios Couris

**Affiliations:** 1Department of Physics, University of Patras, 26504 Patras, Greece; n.chazapis@iceht.forth.gr (N.C.); m.stavrou@iceht.forth.gr (M.S.); 2Institute of Chemical Engineering Sciences (ICE-HT), Foundation for Research and Technology-Hellas (FORTH), 26504 Patras, Greece; 3Department of Mechanical and Manufacturing Engineering, University of Cyprus, 1 Panepistimiou Avenue, Aglantzia, Nicosia 2109, Cyprus; papaparaskeva.georgia@ucy.ac.cy; 4National Institute R&D of Isotopic and Molecular Technologies, 400293 Cluj-Napoca, Romania; alexander.bunge@itim-cj.ro (A.B.); rodica.turcu@itim-cj.ro (R.T.)

**Keywords:** Ir nanoparticles, IrO_2_ nanoparticles, colloidal nanohybrids, z-scan, nonlinear optical response

## Abstract

The present work reports on the synthesis and characterization of iridium (Ir)-based nanohybrids with variable chemical compositions. More specifically, highly stable polyvinylpyrrolidone (PVP) nanohybrids of the PVP-IrO_2_ and PVP-Ir/IrO_2_ types, as well as non-coated Ir/IrO_2_ nanoparticles, are synthesized using different synthetic protocols and characterized in terms of their chemical composition and morphology via X-ray photoelectron spectroscopy (XPS) and scanning transmission electron microscopy (STEM), respectively. Furthermore, their nonlinear optical (NLO) response and optical limiting (OL) efficiency are studied by means of the Z-scan technique, employing 4 ns laser pulses at 532 and 1064 nm. The results demonstrate that the PVP-Ir/IrO_2_ and Ir/IrO_2_ systems exhibit exceptional OL performance, while PVP-IrO_2_ presents very strong saturable absorption (SA) behavior, indicating that the present Ir-based nanohybrids could be strong competitors to other nanostructured materials for photonic and optoelectronic applications. In addition, the findings denote that the variation in the content of IrO_2_ nanoparticles by using different synthetic pathways significantly affects the NLO response of the studied Ir-based nanohybrids, suggesting that the choice of the appropriate synthetic method could lead to tailor-made NLO properties for specific applications in photonics and optoelectronics.

## 1. Introduction

Nanomaterials based on precious metals such as palladium (Pd), platinum (Pt), gold (Au), silver (Ag), rhodium (Rh), ruthenium (Ru) and iridium (Ir) have attracted great attention in recent years due to their unique optical, catalytic and electronic properties. Consequently, such nanomaterials are extensively explored in many diverse areas including catalysis [1,2,3], sensing and biosensing [4,5], biomedicine [6,7], wastewater treatment [8] and optoelectronics [9,10]. 

From the aforementioned precious nanomaterials, not as much work has been reported on Ir-based nanosystems compared to other types of precious metals. The fact that Ir-based nanomaterials are usually characterized by extremely small nanoparticle/cluster diameters makes their characterization quite challenging [11]. Moreover, there is still a knowledge gap in respect to their formation mechanism governing their morphological characteristics and chemical composition [12,13].

Various synthetic approaches have been employed in the preparation of Ir-based nanomaterials [14]. These include chemical, photochemical and electrochemical reduction processes; thermal decomposition; and hydrothermal, sonochemical microwave-assisted synthetic strategies. In most of these synthetic routes, stabilization was accomplished in the presence of polymers and surfactants [15,16], whereas surfactant-free stabilization was also proven to be highly effective, enabling the generation of stable Ir-based nanocolloids [17,18]. Among others, it has been demonstrated that such stabilizing mechanisms may enhance the activity of Ir-based nanomaterials by preventing their agglomeration, while retaining their chemical characteristics [19]. 

Based on the above, it becomes obvious that the scientific community has been highly active with regard to the development of Ir-based nanosystems since the latter are considered to be extremely promising in many applications [11], including electrochemical water splitting [20], wastewater treatment [21], sensing and biosensing [22], biomedicine [23] and optics [24].

The large optical nonlinearities of metal and semiconducting nanoparticles are undoubtedly a key advantage for their implementation in photonic devices [25,26,27,28,29,30,31,32,33,34,35,36,37,38]. To date, immense research efforts have been devoted to the investigation of the exceptional saturable absorption (SA) properties (nonlinear absorption coefficient β, modulation depth α_s_ and saturable intensity I_s_) of low-dimensional nanostructures under ns laser radiation, demonstrating their great prospects for the generation of short laser pulses in Q-switched laser systems [25,26,27,28,29,30,31,32]. Other published research articles, primarily focused on the nonlinear optical (NLO) absorption of semiconducting nanoparticles, have revealed their high optical limiting (OL) efficiency [33,34,35,36,37,38]. Thus, they could be used for OL-related applications, such as the protection of human retinal and sensitive optical devices from an intense laser beam and in several technologically important fields, ranging from medical applications, optical imaging and telecommunications to laser material processing and defense systems [39,40,41,42]. Although numerous studies have appeared so far discussing the NLO response (NLO absorption and refraction) [25,26,27,28,29,30,31,32,43,44,45] and OL efficiency [33,34,35,36,37,38] of various metal and semiconducting nanoparticles, to the best of our knowledge, related studies on Ir-based nanohybrids are rather scarce, narrowing their practical application in photonics and optoelectronics. 

In that context, solutions of Ir-based nanocolloids, namely polyvinylpyrrolidone-stabilized IrO_2_ nanoparticles (PVP-IrO_2_), a mixture of PVP-stabilized Ir/IrO_2_ nanoparticles with an Ir/IrO_2_ ratio of ~0.25 (PVP-Ir/IrO_2_) and a mixture of Ir/IrO_2_ nanoparticles without polymeric coating and an Ir/ IrO_2_ ratio of ~1 (Ir/IrO_2_), were generated by following three different synthetic protocols. PVP was used as a macromolecular steric stabilizer due to its non-toxicity, low cost and high effectiveness in acting as a stabilizing agent for numerous types of inorganic nanomaterials [46]. Furthermore, based on a previous study, it has been demonstrated that PVP-stabilized Ir NPs generated in methanol were characterized by very small diameters while retaining excellent stability in solution for several months [15]. The aforementioned Ir-based nanocolloids were further characterized in respect to their NLO response and OL performances using 4 ns laser pulses at 532 and 1064 nm.

## 2. Materials and Methods

### 2.1. Materials

Iridium (III) chloride hydrate (IrCl_3_·3H_2_O; Mw = 298.58 gmol^−1^, anhydrous basis, 99.9% trace metal basis) and polyvinylpyrrolidone (PVP; average Mw = 1,300,000 gmol^−1^) were purchased from Sigma-Aldrich Chemie GmbH, Tauflirchen, Germany. Sodium hydroxide (NaOH; Mw = 40.00 gmol^−1^) and methanol (MeOH; analytical grade) were purchased from Sigma-Aldrich Chemie GmbH, Tauflirchen, Germany and Sharlau Chemicals (Barcelona, Spain), respectively. All chemicals were used as received from the manufacturer without further purification. The Millex^®^-HP 13 mm filter unit (PES 0.45 μm membrane) that was employed in solution filtration was purchased from Sigma-Aldrich Chemie GmbH, Tauflirchen, Germany.

### 2.2. Synthesis of Ir-Based NP Solutions

Ir-based NP solutions were prepared in methanol, following 3 different experimental protocols (denoted as I, II and III, Figure 1), as described below.

(I) (Ir/IrO_2_) A dispersion of IrCl_3_·3H_2_O (134 mg, 0.45 mmol) was prepared in methanol (5 mL) via vigorous stirring (1250 rpm) for 1 h at room temperature. NaOH (153 mg, 765 mM) dissolved in methanol (5 mL) was then added dropwise (1 mL/min) into the IrCl_3_·3H_2_O/methanol dispersion under stirring conditions (500 rpm). The latter was left to stir for additional 2 h under reflux (65 °C) upon stirring (500 rpm), resulting in a clear, black-colored suspension (Figure 1). This was filtered (using PES 0.45 μm membrane syringe filters), and its color remained stable, while no precipitation phenomena were observed.

(II) (PVP-IrO_2_) PVP (1 g, 9 mmol of vinylpyrrolidone unit, 5 mL) was placed in a 50 mL round-bottom flask, and methanol (5 mL) was added. The resulting mixture was left to stir (500 rpm) for 10 min at room temperature, resulting in a clear, transparent polymer solution. In the meantime, a dispersion of IrCl_3_·3H_2_O (134 mg, 0.45 mmol) was prepared in methanol (5 mL) via vigorous stirring (1250 rpm) for 1 h at room temperature. The latter was added dropwise (1 mL/min) into the methanol solution containing PVP, followed by overnight stirring (500 rpm) at room temperature. The obtained solution was further stirred (500 rpm) for 6 h under reflux at 65 °C to yield a clear, bronze-colored suspension (Figure 1). This was filtered (using PES 0.45 μm membrane syringe filters), and its color remained stable, while no precipitation phenomena were observed.

(III) (PVP-Ir/IrO_2_) PVP (1 g, 9 mmol of vinylpyrrolidone unit, 5 mL) was placed in a 50 mL round-bottom flask and dissolved in a NaOH/methanol solution (153 mg, 765 mM, 5 mL). The resulting mixture was left to stir (500 rpm) for 10 min at room temperature, resulting in a clear, transparent polymer solution. In the meantime, a dispersion of IrCl_3_·3H_2_O (134 mg, 0.45 mmol) was prepared in methanol (5 mL) via vigorous stirring (1250 rpm) for 1 h at room temperature. The latter was added dropwise (1 mL/min) into the NaOH/methanol solution containing PVP, followed by overnight stirring (500 rpm) at room temperature. The obtained solution was further stirred (500 rpm) for 6 h under reflux at 65 °C, resulting in a clear, black-colored suspension (Figure 1). This was filtered (using PES 0.45 μm membrane syringe filters), and its color remained stable, while no precipitation phenomena were observed.

All the above-described Ir-based systems were subjected to 10× dilution prior to their characterization. 

### 2.3. Materials Characterization

The surface chemical composition of the samples was investigated by X-ray photoelectron spectroscopy (XPS) using an XPS spectrometer SPECS equipped with a dual-anode X-ray source Al/Mg, a PHOIBOS 150 2D CCD hemispherical energy analyzer and a multi-channeltron detector with vacuum pressure maintained at 1 × 10^−9^ torr. The particle suspension was dried on an indium foil prior to the XPS measurements. Data analysis and curve fitting were performed using CasaXPS software.

The size and shape of the Ir-based nanostructures were examined by scanning transmission electron microscopy (STEM) with a Hitachi HD2700 equipped with a cold field emission gun, Dual EDX System (X-Max N100TLE Silicon Drift Detector (SDD)) from Oxford Instruments. For the analysis, the methanolic suspension of Ir-based nanoparticles was first dried, and the sample was resuspended in water. The nanoparticles were then centrifuged in an ultracentrifuge for several hours before being resuspended in water. The centrifugation was repeated again, and a suspension of the samples was sonicated (<10 s) with a UP100H ultrasound finger and deposited using the droplet method on a 400-mesh copper grid coated with a thin carbon layer. For both types of analysis, the nominal operating tension was 200 kV. The size of the nanoparticles was determined using the ImageJ software.

### 2.4. Z-Scan Measurements 

The third-order nonlinear optical (NLO) response of the studied Ir-based nanohybrids was investigated by means of the conventional single-beam Z-scan technique. Further information on this technique and the procedures followed for the analysis of the obtained data are reported in detail elsewhere [47], and only a short description is given herein. The Z-scan technique simultaneously provides information about the sign and the magnitude of the NLO absorption and refraction of a sample, which can be expressed in terms of the nonlinear absorption coefficient β and nonlinear refractive index parameter γ′, respectively. These NLO quantities can be determined by measuring the variation in the sample’s normalized transmittance, as it is driven by a stepper motor along the propagation direction (e.g., the z-axis) of a focus laser beam, by means of two different experimental configurations, the so-called “open aperture” (OA) and “closed aperture” (CA) Z-scans. In the former configuration (i.e., the OA Z-scan), the transmitted laser beam is totally collected by a lens and measured by a detector (e.g., a photomultiplier (PMT)), allowing for the determination of the nonlinear absorption coefficient β. Simultaneously, in the latter configuration (i.e., the CA Z-scan), where only a part of the transmitted laser beam is measured by a second PMT after it has passed through a narrow pinhole positioned in the far field, information about the nonlinear refraction of the sample (i.e., the nonlinear refractive index parameter γ′) is provided. 

In general, the presence of a transmission maximum/minimum in the OA Z-scan reveals saturable absorption (SA)/reverse saturable absorption (RSA) behavior of the sample, corresponding to a negative/positive nonlinear absorption coefficient β. Similarly, the presence of a pre-focal transmission minimum followed by a post-focal maximum or vice versa in the CA Z-scan denotes a self-focusing (γ′ > 0) or self-defocusing (γ′ < 0) behavior, respectively. It is interesting to note that the CA Z-scan measurements can be also affected by the NLO absorption of the sample. In this case, the CA Z-scan recordings are asymmetric with respect to the focal plane. Thus, in order to remove the contribution of NLO absorption on the CA Z-scan recording, the latter is divided by the corresponding OA Z-scan, providing the so-called “divided” Z-scan recording, which is used for the direct determination of the nonlinear refractive index parameter γ′.

The values of β and γ′ are calculated by fitting the experimental OA and “Divided” Z-scans with Equations (1) and (2), respectively:(1)$$T(x)=\frac{1}{\sqrt{\pi }(\frac{\beta I_{o}L_{eff}}{1+{\left(x\right)}^{2}})}\int_{-\infty }^{+\infty }\ln\left[1+\frac{\beta I_{o}L_{eff}}{1+{\left(x\right)}^{2}}e^{-t^{2}}\right]dt$$
(2)$$T\left(x\right)=1-\frac{4\gamma \prime kI_{0}L_{eff}x}{\left(1+x^{2})(9+x^{2}\right)}$$
where x = z/z_0_ with z and z_0_ denoting the sample’s position and Rayleigh length, respectively, I_0_ is the on-axis peak irradiance, L_eff_ is the sample’s effective length and k is the wavenumber at the excitation wavelength. 

Using the determined values of β and γ′, the real and the imaginary parts of the third-order susceptibility χ^(3)^ can be deduced through Equations (3) and (4), respectively:(3)$$Im\chi^{\left(3\right)}\left(esu\right)=10^{-7}\frac{c^{2}n_{0}^{2}\beta }{96\pi^{2}\omega }$$
(4)$$Re\chi^{\left(3\right)}\left(esu\right)=10^{-6}\frac{cn_{0}^{2}\gamma \prime }{480\pi^{2}}$$
where c is the speed of light, $n_{0}$ is the refractive index and ω is the frequency of the incident laser beam. To facilitate comparisons among samples having different linear absorption α_0_, the determined NLO parameters (i.e., imaginary/real parts of the third-order nonlinear susceptibility χ^(3)^) are divided by the respective absorption α_0_, providing a figure of merit (FOM) for these quantities.

The laser source used for the Z-scan experiments was a Q-switched Nd:YAG laser system (EKSPLA NT 342/3/UVE/AW), delivering 4 ns laser pulses at 1064 (fundamental frequency) and 532 nm (SHG) with a variable repetition rate from 1 to 10 Hz. For the Z-scan measurements, two different concentrations of the studied Ir-based colloidal solutions in methanol (i.e., 0.5 and 1% *w*/*v*) were prepared and placed into 1 mm path length quartz cells. The laser beam was focused into the samples by means of a 20 cm focal length quartz lens. The laser beam radii at the focal plane were measured using a CCD camera and found to be about (18 ± 2) and (30 ± 2) μm for the 532 and 1064 nm laser outputs, respectively.

### 2.5. Optical Limiting

To assess the optical limiting (OL) performance of the studied Ir-based nanohybrids, under 4 ns 532 nm laser irradiation, the OA Z-scan recordings were used to plot the variation in the sample’s transmittance as a function of the input laser fluence F_in_(z). For the calculation of the values of F_in_(z) at each z-position, the following equation was used:(5)$$F_{in}(z)=\frac{4\sqrt{ln2}E_{in}}{\pi^{3/2}{r(z)}^{2}}$$
where E_in_ is the incident laser energy and r(z) is the beam radius at each z-position, the latter given by the relation:(6)$$r\left(z\right)=r\left(0\right){\left[1+{\left(z/z_{0}\right)}^{2}\right]}^{\frac{1}{2}}$$

To check for possible contribution of nonlinear scattering (NLS) signal to the OL performance of the samples, a high-precision photodiode was used, which was placed on a goniometric stage, monitoring the scattered laser radiation at different angles with respect to the laser propagation. However, within the range of incident laser intensities used for the Z-scan measurements, no significant NLS signal was detected, indicating a rather negligible contribution of NLS to the OL performance of the studied Ir-based nanohybrids.

## 3. Results and Discussion

### 3.1. Chemical Composition and Morphology

STEM was performed on samples I–III (Figure 2). For sample I (Ir/IrO_2_), the nanoparticles could only be seen clustered into larger agglomerates overlapping each other (see also Appendix A), unlike sample III (PVP-Ir/IrO_2_), where at least parts of the sample were found in almost perfect monolayers. The individual nanoparticles were uniform and monodisperse, with an average diameter of 1.6 ± 0.28 nm. The presence of Ir in EDX (Figure 3) confirmed that Ir-containing nanoparticles were generated in all three cases. In the electron micrographs of sample III (PVP-Ir/IrO_2_), the nanoparticles were much more spread out. The average diameter was slightly smaller, at 1.4 ± 0.24 nm, and exhibited uniformity and good dispersity (i.e., fairly monodisperse). The comparatively smaller amount of iridium relative to the amount of carbon (with small contributions from PVP and a large contribution from the carbon grid underground) visible in EDX stems from the fact that the nanoparticles are very small and arranged in a monolayer on the carbon grid. The decrease in tendency to agglomerate is likely due to the steric stabilization provided by PVP, which also resulted in a higher stability in suspension—even repeated washing after centrifugation could not completely remove the polymeric coating. In sample II (PVP-IrO_2_), the separation of nanoparticles by centrifugation was unsuccessful. As a result, the presence of a large amount of the polymer coating prevented the acquisition of clear images of the Ir-based nanoparticles, and hence the nanoparticles shown in Figure 2 (top) are indistinguishable. Nevertheless, the fact that this area does contain nanoparticles is proven by EDX, which clearly shows the presence of Ir, O and N (with the latter verifying the presence of PVP). The comparatively large ratio of Ir to C in EDX makes it likely that also in this case the nanoparticles are stacked in more than a monolayer, similar to sample I.

The XPS investigation of samples II (PVP-IrO_2_,), III (PVP-Ir/IrO_2_) and I (Ir/IrO_2_) evidence differences in their chemical composition. The survey spectra for the three samples are given in Figure 4, while in Figure 5, Figure 6 and Figure 7, the high-resolution XPS spectra of the elements for the above-mentioned systems are provided. As seen in Figure 4, the characteristic signal corresponding to nitrogen appears only in samples II and III as expected, since PVP is present only in those two samples.

According to the XPS spectra appearing in Figure 6, sample II contains only IrO_2,_ because the Ir4f spectrum contains two components that have the corresponding binding energies for Ir^4+^. In the case of sample III, the recorded Ir4f spectrum appearing in Figure 7 shows that Ir^0^ and IrO_2_ co-exist in an atomic concentration ratio Ir^0^/Ir^4+^ = 0.25. Finally, in sample I (Figure 5), a mixture of Ir^0^ and IrO_2_ is also present, with an atomic concentration ratio: Ir^0^/Ir^4+^ = 1.

### 3.2. NLO Properties

In Figure 8, representative UV-Vis-NIR absorption spectra of the prepared Ir-based nanohybrid solutions (PVP-IrO_2_—sample II; PVP-Ir/IrO_2_—sample III; Ir/IrO_2_—sample I) prepared in methanol (0.5% *w*/*v*) are presented. As shown, all the Ir-based nanohybrids exhibited an absorption maximum at 300 nm, ascribed to the Ir-monomer precursor [48], while the absorption spectrum of PVP-IrO_2_ also displayed three shoulders at ~330, 380 nm and 540 nm. More specifically, the absorbance peaks of PVP-IrO_2_ at ~330 and 380 nm are most likely attributed to the iridium (III) chloride salt [49], while the appearance of the absorbance peak at ~540 nm, ascribed to Ir^3+^ to Ir^4+^ transitions, is typical of IrO_2_ nanoparticles [50]. The UV-Vis-NIR absorption spectra shown in Appendix A, which have been taken for all samples in different periods, support the stability of the solutions. As can be seen from this figure, all the measured spectra did not present any change in the morphology and the absorbance values, confirming the high stability of the solutions over a long period of time.

In Figure 9a,c,e, representative OA Z-scans of the prepared Ir-based nanohybrid solutions, obtained under 4 ns 532 nm laser excitation using different laser intensities, are shown. To facilitate comparisons, all measured OA Z-scans correspond to solutions exhibiting the same linear absorption coefficient α_0_, i.e., of about 1.8 cm^−1^, at 532 nm. The solid lines correspond to the best fits of the experimental data points (solid symbols) according to Equation (1). Similar OA Z-scan measurements performed under 4 ns 1064 nm laser excitation revealed insignificant NLO absorption for all samples. In addition, the possible contribution of the solvent (i.e., methanol) and polymer (i.e., PVP) to the NLO absorptive response of the solutions was checked and found to be insignificant within the range of laser intensities used (i.e., up to 180 MW/cm^2^) for the Z-scan measurements. Therefore, the presented OA Z-scans of Figure 9a,c,e provide the NLO absorptive response of the Ir-based nanohybrids directly. As can be seen from Figure 9a, the OA Z-scans of PVP-IrO_2_ revealed a maximum at the focal plane (i.e., at z = 0), indicative of a typical saturable absorption (SA, β < 0) behavior. In addition, this transmission maximum was found to increase with incident laser intensity, retaining, however, a clear transmission maximum even for the highest laser intensity used, implying that a one-photon absorption (1PA)-related nonlinearity occurs under 532 nm laser excitation. However, since the bandgap energy of the studied IrO_2_ semiconducting nanoparticles, evaluated to be ~2.7 eV (see in Appendix A), is larger than the corresponding excitation photon energy at 532 nm (~2.33 eV), the SA behavior of PVP-IrO_2_ may be assisted by the existence of abundant defect states within the bandgap, generated from oxygen-deficient phases [51,52]. In particular, due to the presence of defect states, the energy bandgap of IrO_2_ can be bridged by electrons absorbing photons with lower energy, leading to a progressive increase in the carriers accumulated in its conduction band. When the sample is close to the focal plane (z = 0), where the laser intensity is sufficiently strong, the interband optical transitions become more efficient, resulting in the depletion of all the available states in the conduction band of IrO_2_. Consequently, any further excitation of electrons is blocked due to the Pauli exclusion principle, giving rise to the appearance of SA behavior (β < 0). From the fitting of the OA Z-scans of PVP-IrO_2_ presented in Figure 9a, the mean value of the nonlinear absorption coefficient β was determined to be (−115.6 ± 18.0) × 10^−11^ m/W.

Besides their large values of the nonlinear absorption coefficient β, materials suitable for saturable absorption applications are characterized by high modulation depth α_s_ and low saturable intensity I_sat_. For the determination of the α_s_ and I_sat_ values of PVP-IrO_2_, the experimentally obtained values of normalized transmittance (solid points) have been plotted as a function of the incident laser intensity (see Figure 9b) and fitted using Equation (7): (7)$$T=1-\left(\frac{\alpha_{s}}{1+I/I_{sat}}+\alpha_{ns}\right)$$
where α_ns_ represents the non-saturable components. From the fitting of intensity-dependent transmittance, the values of α_s_ and I_sat_ were determined to be ~4% and 80 MW/cm^2^, respectively. In order to highlight the efficiency of the PVP-IrO_2_ system as a saturable absorber for the generation of laser pulses in Q-switched laser systems, its saturable absorption properties (i.e., the nonlinear absorption coefficient β (or Imχ^(3)^), the modulation depth α_s_ and the saturable intensity I_sat_) have been compared to those of other metal and semiconducting nanoparticles reported to exhibit exceptional SA response, including silver (Ag), gold (Au), platinum (Pt), palladium (Pd), zinc oxide (ZnO), vanadium pentoxide (V_2_O_5_), iron oxide (γ-Fe_2_O_3_), copper oxide (CuO) and copper hydroxide (Cu(OH)_2_) [25,26,27,28,29,30,31,32]. As presented in Table 1, the figure of merit of Imχ^(3)^ (i.e., the quantity Imχ^(3)^/α_0_) as well as the values of α_s_ and I_sat_ of PVP-IrO_2_ are comparable to those of other nanomaterials, emphasizing the efficiency of the studied IrO_2_ nanoparticles for saturable absorption applications. 

As shown in Figure 9c,e, the measured OA Z-scans of the solutions containing a mixture of Ir and IrO_2_ nanoparticles, i.e., the PVP-Ir/IrO_2_ and Ir/IrO_2_ nanohybrid solutions, presented a deep focus, increasing with the laser pulse intensity, implying a positive absorptive nonlinearity (i.e., RSA behavior, β > 0). From the fitting of these OA Z-scans with Equation (1), the average values of the nonlinear absorption coefficient β of PVP-Ir/IrO_2_ and Ir/IrO_2_ were determined to be (13.8 ± 2.0) × 10^−11^ and (24.2 ± 2.0) × 10^−11^ m/W, respectively, suggesting that the increase in the IrO_2_ content leads to significant enhancement of the NLO absorption of these nanohybrids. The same conclusion can also be drawn from the comparison of the β value of the PVP-IrO_2_ solution, containing only IrO_2_ nanoparticles, with the corresponding values of PVP-Ir/IrO_2_ and Ir/IrO_2_. In particular, as mentioned above, the β value of PVP-IrO_2_ was found to be (−115.6 ± 18.0) × 10^−11^ m/W, that is, ~5 and 10 times larger than those of Ir/IrO_2_ and PVP-Ir/IrO_2_, respectively.

In general, the RSA behavior of the studied nanostructures can be explained in terms of different physical processes, such as excited state absorption (ESA), two-photon absorption (2PA), nonlinear scattering (NLS), and thermal cumulative effects [53]. The presence of NLS was investigated and found to be negligible under the present experimental conditions. In addition, any contribution arising from thermal effects should be excluded, as the laser repetition rate was set at as low as 1 Hz. Furthermore, trials using higher repetition rates (i.e., up to 10 Hz) did not show significant modifications of the sample’s transmittance, ensuring the absence of such effects. Consequently, the NLO absorptive response of the present nanoparticles can be most likely attributed to ESA and/or 2PA processes. For a better understanding of the physical processes responsible for the NLO absorptive response of the PVP-Ir/IrO_2_ and Ir/IrO_2_ solutions, it should be considered that the optical bandgap of the IrO_2_ semiconducting nanoparticles, estimated for the sample containing only IrO_2_ nanoparticles (PVP-IrO_2_), is about 2.7 eV (see Appendix A) and that Ir nanoparticles exhibit zero bandgap energy. Therefore, since the bandgap of IrO_2_ nanoparticles lies above the sum of energy of two photons at 532 nm (~2.33 eV), only 2PA can occur in IrO_2_. On the other hand, the zero bandgap of Ir nanoparticles facilitates the manifestation of ESA under 532 nm laser excitation. However, as discussed above, the irradiation of PVP-IrO_2_ nanohybrids at 532 nm leads to SA behavior. As a result, it is reasonable to assume that the photo-response mechanism causing the NLO absorption of the present nanohybrids can be assigned to ESA due to the presence of Ir nanoparticles.

Materials exhibiting strong RSA behavior are promising candidates for OL-related applications, such as the protection of human retinal and delicate optical sensors from high-power laser radiation. These materials should exhibit high transmittance, obeying the linear Beer–Lambert law, for low incident laser intensities or fluences and a decreasing transmittance for higher incident fluences. The value of input fluence at which the transmittance begins to deviate from the Beer–Lambert regime, defined as the optical limiting onset (OL_on_), is usually used for the assessment of the OL efficiency of a material. 

For the determination of the OL_on_ values of the PVP-Ir/IrO_2_ and Ir/IrO_2_ nanohybrids, their transmittance at each laser radiation intensity was evaluated from the OA Z-scan transmittance data shown in Figure 9d,f using Equations (5) and (6). The obtained results are presented in Figure 9d,f, where, as can be seen, the OL_on_ values of PVP-Ir/IrO_2_ and Ir/IrO_2_ are approximately 0.38 and 0.18 J/cm^2^, respectively, while their normalized transmittance drops at 78 and 75% for the highest incident laser fluence. Since ESA is the physical process responsible for the RSA behavior of the studied Ir-based nanohybrids, their OL performance can be described in terms of this mechanism.

For a more comprehensive picture of the OL efficiency of PVP-Ir/IrO_2_ and Ir/IrO_2_, the determined values of β (or Imχ^(3)^) and OL_on_ have been compared to those of other semiconducting nanoparticles reported as efficient OL materials, such as zinc oxide (ZnO), indium zinc oxide (InZnO), titanium dioxide (TiO_2_), nickel oxide (NiO), chromium oxide (Cr_2_O_3_), tungsten trioxide (WO_3_), antimony selenide (Sb_2_Se_3_), cadmium sulfide (CdS) and silver sulfide (Ag_2_S) [33,34,35,36,37,38]. The results, obtained under similar excitation conditions (i.e., pulse duration and incident wavelength), are listed in Table 2. As can be seen, the present samples revealed comparable FOM and OL_on_ values to those corresponding to other nanostructured materials included in Table 2, indicating their efficiency for OL-related applications.

In Figure 10, representative “Divided” Z-scans of the PVP-IrO_2_, PVP-Ir/IrO_2_ and Ir/ IrO_2_ colloidal solutions prepared in methanol, obtained using 4 ns laser pulses at 532 (Figure 10a) and 1064 nm (Figure 10b), are shown. To facilitate comparisons, the measured “Divided” Z-scans correspond to solutions exhibiting a linear absorption coefficients α_0_ of about 1.8 and 0.4 cm^−1^ at 532 and 1064 nm, respectively. The continuous lines represent the best-fitting curve to the obtained experimental data (solid points) according to Equation (2). As can be seen, all solutions displayed a peak–valley configuration under both excitation wavelengths, denoting a self-defocusing behavior (i.e., γ′ < 0). Moreover, PVP and methanol did not exhibit measurable NLO refraction under the present excitation conditions. Therefore, the nonlinear refractive index parameter γ′ of the studied nanohybrids can be straightforwardly obtained from the “Divided” Z-scans shown in Figure 10. Consequently, from the fitting of these recordings, the nonlinear refractive index parameter γ′ of PVP-IrO_2_, PVP-Ir/IrO_2_ and Ir/IrO_2_ were determined to be about (−89.8 ± 16.0) × 10^−21^, (−20.0 ± 2.0) × 10^−21^ and (−28.8 ± 2.0) × 10^−21^ m^2^/W, respectively, at 532 nm, and about (−28 ± 4) × 10^−21^, (−7.8 ± 0.4) × 10^−21^, (−13.3 ± 2.0) × 10^−21^ m^2^/W at 1064 nm. The above values demonstrate nicely that the increase in IrO_2_ concentration results in the enhancement of the NLO refraction of the samples.

At this point, it is interesting to note that during ns laser excitation, free carrier refraction and thermal lensing are the most sizable contributions to the observed refractive nonlinearity. The former process is attributed to the more efficient electronic transitions that occur, while the latter is due to the increase in the thermal energy accumulated in the sample [54,55]. However, in the present case, the manifestation of thermal lensing should be excluded, since the laser repetition rate was kept at as low as 1 Hz, while measurements using higher repetition rates (i.e., up to 10 Hz) did not show significant modification of the sample’s transmittance. Consequently, it can be safely concluded that free carrier refraction is the dominant mechanism contributing to the NLO refraction of the present nanohybrids. 

By following the standard procedures used for the analysis of the obtained Z-scan recordings [47], the values of NLO parameters, i.e., the nonlinear absorption coefficient β, the nonlinear refractive index parameter γ′ and the third-order susceptibility χ^(3)^, have been calculated and are summarized in Table 3. To facilitate the comparison among the samples with different absorption α_0_, the quantity χ^(3)^/α_0_ is also included in Table 3.

From the χ^(3)/^α_0_ values shown in Table 3, it is concluded that the content of IrO_2_ nanoparticles significantly affects the NLO response of the present Ir-based nanohybrids. More specifically, the NLO response of PVP-IrO_2_, containing only IrO_2_ nanoparticles, was found to be enhanced by factors of ~10 and 5 compared to PVP-Ir/IrO_2_ and Ir/IrO_2_, respectively, for visible (532 nm) laser excitation and by factors of ~4 and 2.5 for infrared (1064 nm) excitation. In addition, among the solutions containing a mixture of Ir and IrO_2_ nanoparticles, Ir/IrO_2_ (Ir/IrO_2_ ratio ~1) revealed ~2 and 1.5 times larger χ^(3)/^α_0_ values than PVP-Ir/IrO_2_ (Ir/IrO_2_ ratio ~0.25) under 532 and 1064 laser excitation, respectively. Therefore, the present findings indicate that the modification of atomic content of Ir-based nanohybrids through the choice of the suitable synthesis method can be used in the tailoring of their NLO response.

In addition, PVP-IrO_2_ revealed saturable absorption (β < 0), while both PVP-Ir/IrO_2_ and Ir/IrO_2_ exhibited reverse saturable absorption (β > 0). This sign alternation of the nonlinear absorption coefficient β indicates that they can be used either as saturable absorbers or optical limiters by modifying the content of IrO_2_ nanoparticles. Concerning the NLO refraction, all the studied nanoparticles exhibited large values of γ′ (i.e., in the order of 10^−17^ m^2^/W) under both excitation wavelengths, which are comparable and even larger than those of other metal and semiconducting nanoparticles (see Appendix A), suggesting that they can be suitable for optical-switching applications. 

## 4. Conclusions

In summary, Ir-based nanocolloids were synthesized in methanol, in the presence and absence of PVP, by following three different synthetic procedures, affording solutions with different ratios of Ir and IrO_2_ nanoparticles. Their NLO response and OL efficiency was investigated under 4 ns laser excitation at 532 and 1064 nm. The obtained results reveal that the choice of different synthetic strategies significantly affects the NLO properties of the Ir-based nanohybrids, especially their NLO absorption. More specifically, the nanocolloidal system containing only IrO_2_ nanoparticles, exhibits strong saturable absorption. On the other hand, highly stable solutions containing a mixture of Ir and IrO_2_ nanoparticles with different ratios reveal very efficient OL performance for visible laser excitation. In addition, it was found that the IrO_2_ content of the solutions has a great impact on their NLO absorptive and refractive response, with the solutions having a higher Ir/IrO_2_ ratio revealing larger values of nonlinear absorption coefficient β and nonlinear refractive index parameter. The findings of the present study demonstrate that the variation in the Ir//IrO_2_ ratio of Ir-based nanohybrid solutions by choosing the appropriate synthetic strategy can result in at-will NLO response in view of specific photonic and optoelectronic applications. 

## Figures and Tables

**Figure 1 nanomaterials-13-02131-f001:**
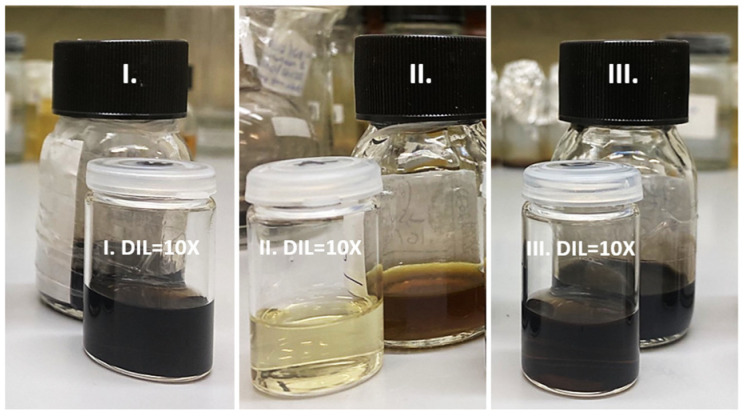
Highly stabilized Ir-based NP methanol solutions prepared by following 3 different synthetic protocols: (**I**) Ir/IrO_2_; (**II**) PVP-IrO_2_; and (**III**) PVP-Ir/IrO_2_.

**Figure 2 nanomaterials-13-02131-f002:**
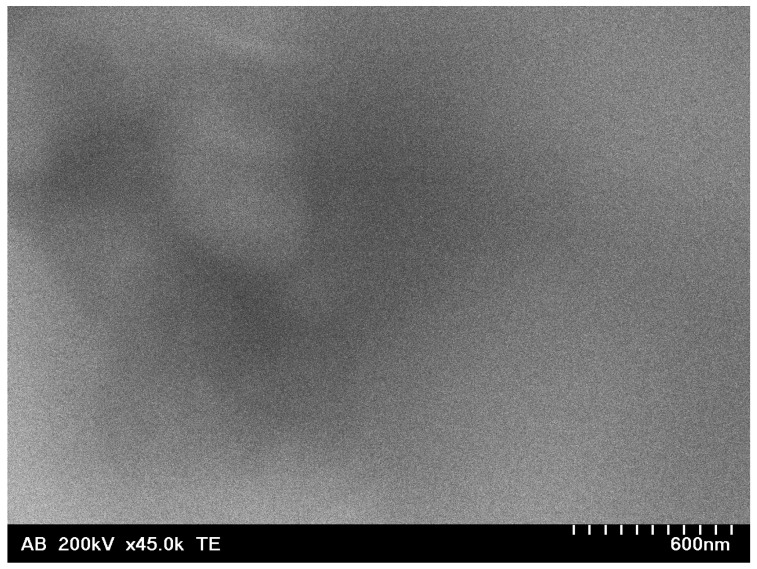
TEM of samples II (PVP-IrO_2_, **top**), III (PVP-Ir/IrO_2_, **middle**) and I (Ir/IrO_2_, **bottom**). Inset: size distributions for samples III and I.

**Figure 3 nanomaterials-13-02131-f003:**
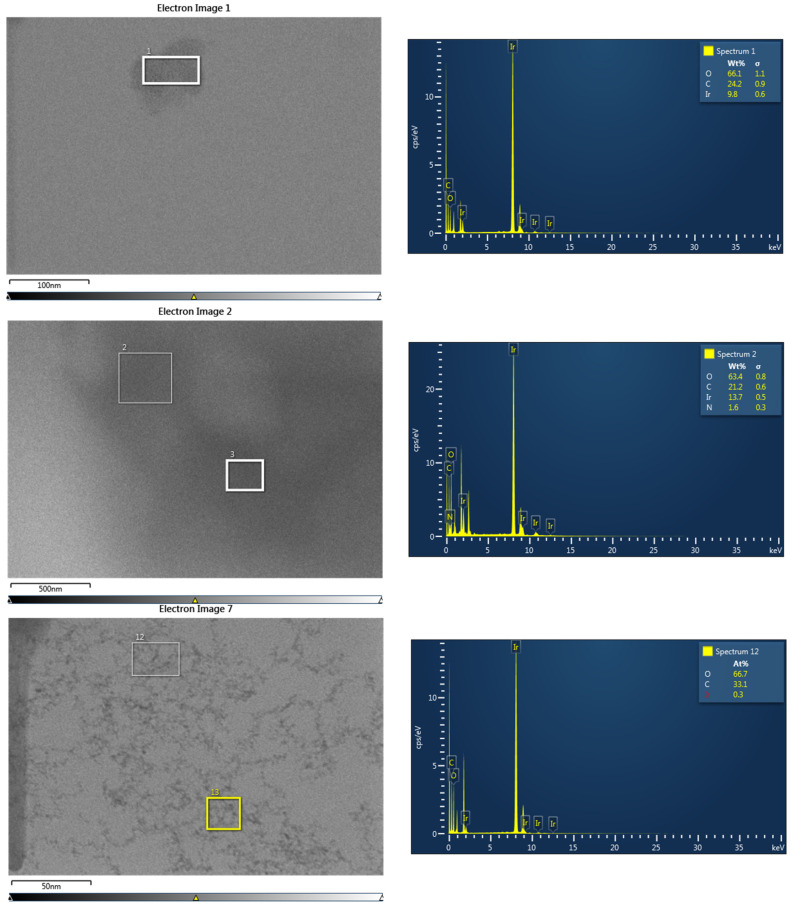
EDX of samples I (Ir/IrO_2_, **top**), II (PVP-IrO_2_, **middle**) and III (PVP-Ir/IrO_2_, **bottom**).

**Figure 4 nanomaterials-13-02131-f004:**
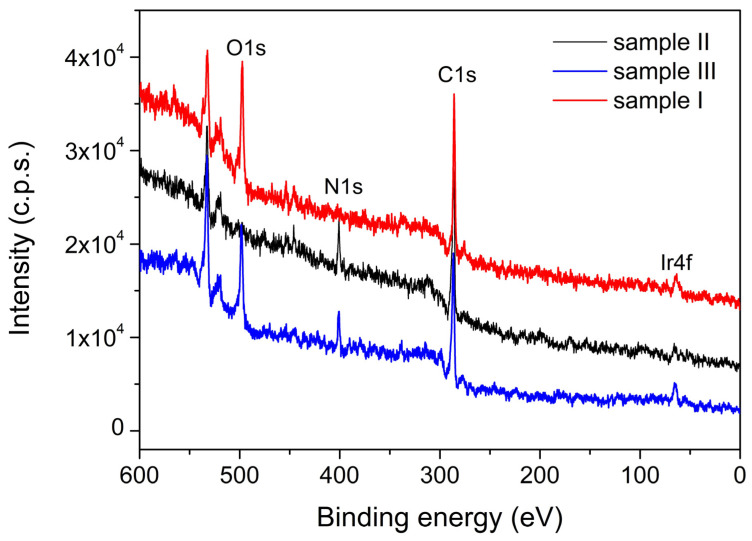
The XPS survey spectra for samples II (PVP-IrO_2_), III (PVP-Ir/IrO_2_) and I (Ir/IrO_2_).

**Figure 5 nanomaterials-13-02131-f005:**
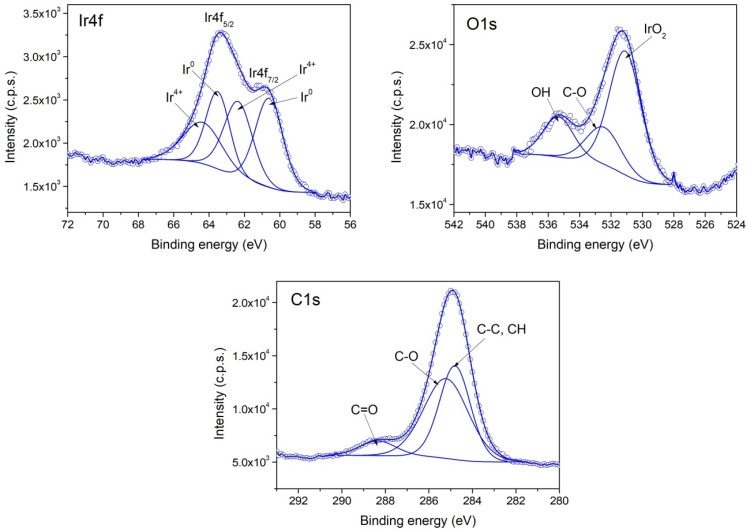
XPS spectra of Ir4f, O1s and C1s core levels from Sample I.

**Figure 6 nanomaterials-13-02131-f006:**
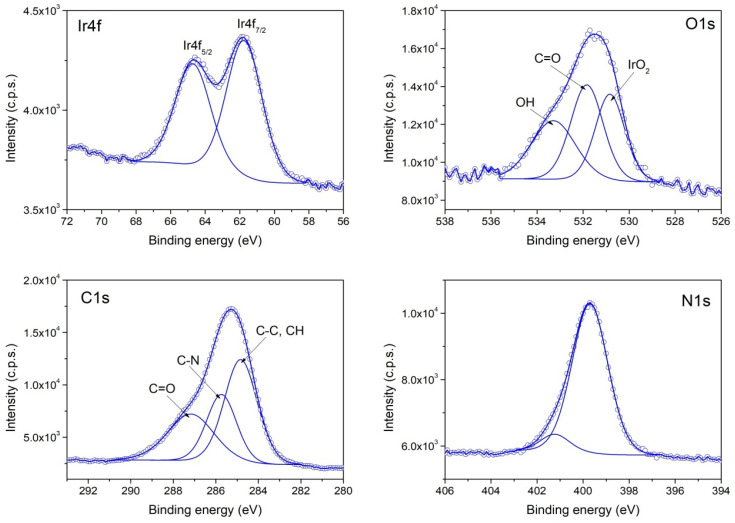
XPS spectra of Ir4f, O1s and C1s and N1s core levels from Sample II.

**Figure 7 nanomaterials-13-02131-f007:**
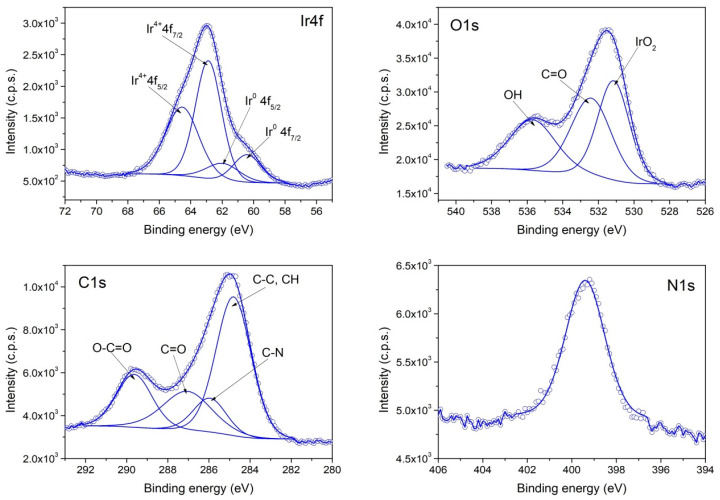
XPS spectra of Ir4f, O1s and C1s and N1s core levels from Sample III.

**Figure 8 nanomaterials-13-02131-f008:**
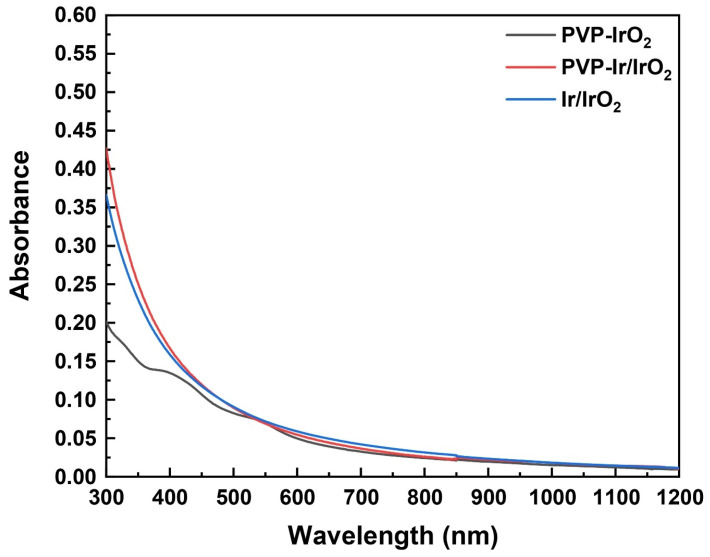
UV-Vis-NIR absorption spectra of Ir-based nanohybrids (all corresponding to a 0.5% *w*/*v* solution concentration).

**Figure 9 nanomaterials-13-02131-f009:**
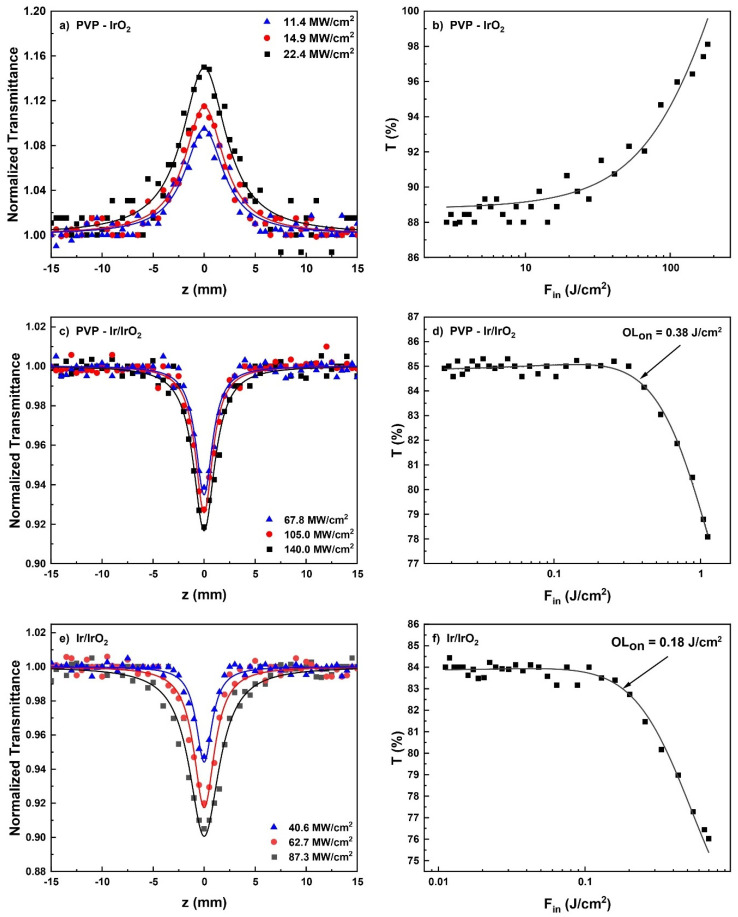
OA Z-scans (**a**,**c**,**e**) and variation in the transmittance (**b**,**d**,**f**) of PVP-IrO_2_, PVP-Ir/IrO_2_, Ir/IrO_2_ nanohybrid solutions, under 4 ns, 532 nm laser excitation, using different laser excitation intensities.

**Figure 10 nanomaterials-13-02131-f010:**
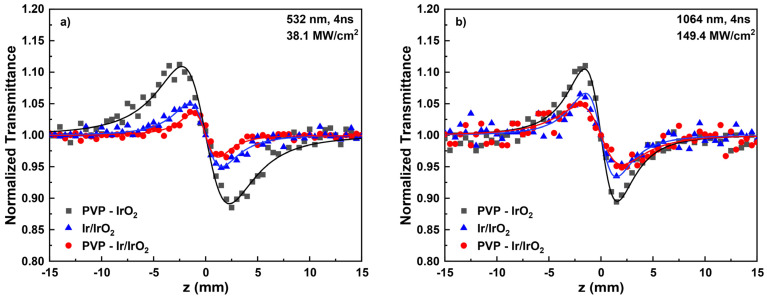
“Divided” Z-scans of PVP-IrO_2_, PVP-Ir/IrO_2_, Ir/IrO_2_ nanohybrid solutions under (**a**) 532 and (**b**) 1064 nm laser excitation.

**Table 1 nanomaterials-13-02131-t001:** Saturable absorption properties of PVP-IrO_2_, Ag, Au, Pt ZnO, V_2_O_5_, Pt, γ-Fe_2_O_3_, Cu(OH)_2_, Cu(OH)_2_/CuO and Pd nanoparticles.

Sample	Excitation Conditions	α_0_ (cm^−1^)	β × 10^−11^ (m/W)	Imχ^(3)^ × 10^−13^ (esu)	Imχ^(3)^/α_0_ × 10^−13^ (esu cm)	α_s_ (%)	I_sat_ (MW/cm^2^)	Ref.
PVP-IrO_2_	4 ns,	1.77	−115.6	−53.0	−30.6	4	80	this work
	532 nm							
Ag	7 ns,	~9.2	−2562	−1178.0	128	N/A	32.1	[25]
	532 nm							
Au	4 ns,	~0.9	−17.6	−8.3	9.2	N/A	230	[26]
	532 nm							
Pt	10 ns,	0.91	−3	−8.53	9.4	N/A	18	[27]
	532 nm							
Pd	4 ns, 532 nm 4 ns, 1064 nm	7.5	−54	−32.5	4.3	N/A	N/A	[28]
		6.8	−5.9	−6.7	0.99	N/A	N/A	
ZnO	400 ns,	N/A	N/A	N/A	N/A	5	60	[29]
	1560 nm							
V_2_O_5_	0.7 ns,	N/A	N/A	N/A	N/A	21.8	0.33	[30]
	1064 nm							
γ-Fe_2_O_3_	4 ns,	~5.1	N/A	−2.2	0.43	N/A	N/A	[31]
	532 nm							
	4 ns,	~1.15	N/A	−1.5	1.3	N/A	N/A	
	1064 nm							
Cu(OH)_2_	4 ns,	1	−97.3	−44.7	−44.7	N/A	N/A	[32]
	532 nm							
Cu(OH)_2_/CuO	4 ns,	1	−594	−273	−273	N/A	N/A	
	532 nm							

**Table 2 nanomaterials-13-02131-t002:** NLO absorption and optical limiting properties of PVP-Ir/IrO_2_, Ir/IrO_2_, ZnO, InZnO, TiO_2_, NiO, Cr_2_O_3_, WO_3_, Sb_2_Se_3_, CdS and Ag_2_S nanoparticles.

Sample	Excitation Conditions	α_0_ (cm^−1^)	β × 10^−11^ (m/W)	Imχ^(3)^ × 10^−13^ (esu)	Imχ^(3)^/α_0_ × 10^−13^ (esu cm)	OL_on_ (J/cm^2^)	Ref.
PVP-Ir/IrO_2_	4 ns, 532 nm	1.8	13.8	6.3	3.5	0.38	this work
Ir/IrO_2_		1.79	24.2	11.1	6.2	0.18	
ZnO	6 ns, 532 nm	2.7 × 10^5^	4.86 × 10^−4^	2.6 × 10^−3^	0.96 × 10^−8^	N/A	[33]
InZnO		1.8 × 10^5^	5.58 × 10^−4^	3.4 × 10^−3^	1.9 × 10^−8^	N/A	
TiO_2_	6 ns, 532 nm	~6.9	395	181.6	26.3	~1	[34]
NiO	5 ns, 532 nm	N/A	3.5–31	1.9–16.5	N/A	~0.8–1	[35]
Cr_2_O_3_	4 ns, 1064	0.31	3.17	3	9.7	~0.8	[36]
WO_3_		0.25	2.51	2.4	9.6	~1.9	
Sb_2_Se_3_	15 ns, 532 nm		50			2	[36]
CdS	4.1 ns, 532 nm	1.37	N/A	N/A	N/A	1.4	[38]
Ag_2_S		0.73	N/A	N/A	N/A	0.6	

**Table 3 nanomaterials-13-02131-t003:** Determined NLO parameters of PVP-stabilized IrO_2_ and Ir/IrO_2_ and non-polymeric Ir/IrO_2_ nanohybrids under 4 ns visible (532 nm) and infrared (1064 nm) laser excitation for different solution concentrations.

λ (nm)	Samples	C (mmol/L)	α_0_ (cm^−1^)	β × 10^−11^ (m/W)	γ′ × 10^−18^ (m^2^/W)	χ^(3)^ × 10^−13^ (esu)	χ^(3)^/α_0_ × 10^−13^ (esu cm)
532 nm	PVP-IrO_2_	4.5	1.77	−115.6 ± 18.0	−89.8 ± 16.0	113.6 ± 20.0	65.3 ± 11.0
		0.6	0.24	−18.2 ± 2.0	−11.1 ± 2.0	15.0 ± 2.0	62.5 ± 10.0
	PVP-Ir/IrO_2_	[Ir]: 0.9; [IrO_2_]: 3.6	2.92	21.0 ± 2.0	−30.5 ± 3.0	35.6 ± 4.0	12.2 ± 1.0
		[Ir]: 0.5; [IrO_2_]: 2.1	1.8	13.8 ± 2.0	−20.0 ± 2.0	23.3 ± 3.0	12.9 ± 1.0
	Ir/IrO_2_	[Ir]: 2.2; [IrO_2_]: 2.2	3.15	47.7 ± 6.0	51.9 ± 6.0	61.4 ± 7.0	19.4 ± 2.0
		[Ir]: 1.2; [IrO_2_]: 1.2	1.79	24.2 ± 2.0	−28.8 ± 2.0	34.2 ± 3.0	19.1 ± 2.0
1064 nm	PVP-IrO_2_	4.5	0.39	-	−28 ± 4	31.9 ± 5.0	82.9 ± 12.0
		2.5	0.22	-	−14.3 ± 1.0	16.3 ± 1.0	75.8 ± 7.0
	PVP-Ir/IrO_2_	[Ir]: 0.9; [IrO_2_]: 3.6	0.52	-	−10.2 ± 1.0	11.3 ± 1.0	21.6 ± 2.0
		[Ir]: 0.6; [IrO_2_]: 2.6	0.38	-	−7.8 ± 0.4	8.7 ± 0.4	22.6 ± 1.0
	Ir/IrO_2_	[Ir]: 2.2; [IrO_2_]: 2.2	0.71	-	−19.8 ± 2.0	21.9 ± 3.0	30.9 ± 4.0
		[Ir]: 1.3; [IrO_2_]: 1.3	0.43	-	−13.3 ± 2.0	14.7 ± 2.0	34.0 ± 4.0

## Data Availability

Not applicable.

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
