# Peer review of "Iridium-Based Nanohybrids: Synthesis, Characterization, Optical Limiting, and Nonlinear Optical Properties"

_nanomaterials, 2023, doi:10.3390/nano13142131_

Round 1

Reviewer 1 Report

This manuscript demonstrates the synthesis of Ir-based nanocolloids  and the optical limiting, and nonlinear optical properties.  Nano

particles are characterized in terms of their chemical composition and morphology by X-ray Photoelectron Spectroscopy (XPS) and Scanning Transmission Electron Microscopy (STEM) respectively.   The PVP-Ir/IrO2 and Ir/IrO2 systems exhibit exceptional OL performance, while PVP-IrO2 presents very strong saturable absorption (SA) behavior. I recommed to accept after a major revision.

1. The authors claimed the "highly stable", please provide the proof.

2. The characterization of the nanohybrids should be further improved. High sulution TEM images should be provided. The images in Figure 2 and Figure 3 are unclear. The particle size distribution needs to be supplemented. 

3. Does particle size affect NLO performance? Can the authors tune the particle size?

4. Does the concentration of the nanobybrids affect the NLO performance? Please indicate the concentration information in Table 2 and Table 3. 

1. The authors claimed the "highly stable", please provide the proof.

2. The characterization of the nanohybrids should be further improved. High sulution TEM images should be provided. The images in Figure 2 and Figure 3 are unclear. The particle size distribution needs to be supplemented. 

3. Does particle size affect NLO performance? Can the authors tune the particle size?

4. Does the concentration of the nanobybrids affect the NLO performance? Please indicate the concentration information in Table 2 and Table 3. 

Author Response

Reviewer 1

Comments and Suggestions for Authors

This manuscript demonstrates the synthesis of Ir-based nanocolloids and the optical limiting, and nonlinear optical properties. Nanoparticles are characterized in terms of their chemical composition and morphology by X-ray Photoelectron Spectroscopy (XPS) and Scanning Transmission Electron Microscopy (STEM) respectively. The PVP-Ir/IrO2 and Ir/IrO2 systems exhibit exceptional OL performance, while PVP-IrO2 presents very strong saturable absorption (SA) behavior. I recommend to accept after a major revision.

Comment 1: “The authors claimed the "highly stable", please provide the proof.”

Authors’ Reply: We thank the reviewer for this comment. With the term “highly stable” we mean that no nanoparticle agglomeration phenomena were observed even after several months. The obtained solutions remained transparent, and no precipitates were detected (see photographs of the obtained solutions provided in Figure 1). In addition, the stability of the dispersions is supported by the UV-Vis-NIR absorption spectra shown in Figure S3, all corresponding to the same concentration solution of 5% w/v, which have been taken for all samples in different periods, i.e., in July and May. As can be seen, all the measured spectra did not present any change in the morphology and the absorbance values, confirming the high stability of the nanohybrids over a long period of time.

Comment 2: “The characterization of the nanohybrids should be further improved. High sulution TEM images should be provided. The images in Figure 2 and Figure 3 are unclear. The particle size distribution needs to be supplemented.

Authors’ Reply: Unfortunately, with the device that we have available the acquirement of high-resolution TEM images is not feasible. Higher resolution is accompanied with a lower amount of contrast, and the contrast was difficult enough for the images that we had to achieve, due to a copious amount of polymer in the solution, which was not possible to remove completely. This is also the reason why the images in Figure 2 and 3 are not very clear. Initial efforts that were unsuccessful, were focused on acquiring TEM images from the suspension without treatment. Only repeated centrifugation (as described in the Methods section) and transfer in water, resulted to some improvement of the image quality. In line with previous literature reports, the nanoparticles were in fact so small that even in an ultracentrifuge, very few of them could be collected at the bottom, while still, a quite large amount of polymer coating was retained. Longer centrifugation times did not lead to a better separation.

In response to the Reviewer’s comment, the particle size for samples I and III (where individual nanoparticles can be discerned) has been added as an inset in Figure 2.

Comment 2: “Does particle size affect NLO performance? Can the authors tune the particle size?”

Authors’ Reply: We thank the reviewer for giving us the opportunity to clarify that point. When the crystallite size is reduced to the order of exciton Bohr radius αΒ, quantum size effects appear and drastic changes in NLO properties are expected. The quantum confinement effect in semiconductor nanocrystals can be classified into two regimes, i.e., the strong and the weak confinement regimes, according to the ratio of nanocrystal radius R to αΒ, i.e., R/αB. [Sreeja, R.; Aneesh, P. M.; Aravind, A.; Reshmi, R.; Philip, R.; Jayaraj, M. K. Size-Dependent Optical Nonlinearity of Au Nanocrystals. J. Electrochem. Soc. 2009, 156, K167.] Nonlinear optical properties in nanocrystals have been investigated for the corresponding confinement regimes. In the strong-confinement regime, the photoexcited electron and hole are individually confined. Under this regime, the size dependence of the NLO response has been studied, but the results are inconsistent; a larger χ(3) value for larger size nanoparticles has been found for CdSxSe1−x nanocrystals by the saturation spectroscopy and degenerate four-wave mixing (DFWM) measurements, while other studies have shown that larger χ(3) values are obtained with decreasing sizes. [Roussignol, Ph.; Ricard, D.; Flytzanis, Chr. Quantum Confinement Mediated Enhancement of Optical Kerr Effect in CdSXSe1-X Semiconductor Microcrystallites. Appl. Phys. B; Photophys. Laser Chem. 1990, 51, 437–442.; Hall, D. W.; Borrelli, N. F. Absorption Saturation in Commercial and Quantum-Confine CdSxSe1−x Doped Glasses. J. Opt. Soc. Am. B 1988, 5, 1650.] In the weak-confinement regime, the Coulomb interaction between the electron and hole yields an exciton which is confined as a quasiparticle. Theoretical studies have shown that the confinement of the excitonic envelope wave function due to the infinite barrier potential gives rise to the enhancement in oscillator strength for an exciton within the nanocrystal by a factor of R3B3, hence χ(3) depends on the crystallite size. The important role of the strong oscillator strength in the size dependent enhancement of nonlinearity has been experimentally shown for CuCl nanocrystals.[Kataoka, T.; Tokizaki, T.; Nakamura, A. Mesoscopic Enhancement of Optical Nonlinearity in Cucl Quantum Dots: Giant-Oscillator-Strength Effect on Confined Excitons. Phys. Rev. B 1993, 48, 2815–2818.] Therefore, it is reasonable to expect that the NLO response of the studied IrO2 nanoparticles strongly depends on the crystallite sizes.

Similarly, there are several investigations reporting that metallic nanoparticles also show size-dependent NLO response. The same behavior has been observed either for on-resonant or off-resonant laser excitation with the surface plasmon of metallic nanoparticles. [Sreeja, R.; Aneesh, P. M.; Aravind, A.; Reshmi, R.; Philip, R.; Jayaraj, M. K. Size-Dependent Optical Nonlinearity of Au Nanocrystals. J. Electrochem. Soc. 2009, 156, K167.; Fu, Y.; Ganeev, R. A.; Krishnendu, P. S.; Zhou, C.; Rao, K. S.; Guo, C. Size-Dependent off-Resonant Nonlinear Optical Properties of Gold Nanoparticles and Demonstration of Efficient Optical Limiting. Opt. Mater. Express 2019, 9, 976.]

However, in the present work, the studied nanoparticles had very similar size distribution and only the effect of Ir and IrO2 content on their NLO response was investigated.

As previously reported, depending on the synthetic methodology employed for the synthesis of Ir-based nanomaterials, the particle size, morphology and chemical composition may be tuned and consequently their physicochemical properties could be also altered [Cui, M.-L.; Chen, Y.-S.;  Xie, Q.-F.;  Yang, D.-P.;  Han, M.-Y. Synthesis, Properties and Applications of Noble Metal Iridium Nanomaterials, Coord. Chem. Rev. 2019, 387, 450-462]. However, as recently reported by J. Quinson [Quinson, J. Iridium and IrOx Nanoparticles: An Overview and Review of Syntheses and Applications, Adv. Colloid Interface Sci. 2022, 303, 102643], Ir NPs exhibit certain challenges when it comes to their size, since very often extremely small size clusters and NPs are easily obtained, however these are very difficult to characterize and understand the different mechanisms involved in their formation.

Comment 4: “Does the concentration of the nanobybrids affect the NLO performance? Please indicate the concentration information in Table 2 and Table 3.”

Authors’ Reply: We thank the reviewer for this comment. For the NLO measurements, different concentration dispersions were prepared, and thus with different concentrations of Ir and IrO2 nanoparticles and were studied. Following reviewer’s suggestion, the corresponding concentrations of the nanoparticles have been added in the manuscript and are listed in Table 3.

The effect of the concentration of the nanohybrids on the NLO response of each sample is reflected in the obtained results which are presented in Table 3. More precisely, from this table, an increasing trend of the NLO parameters is observed with the increase of concentration, denoting that the NLO response of the solutions strongly depends on the concentration of Ir and IrO2 nanoparticles. In addition, as it is stated in the manuscript, the contribution of the polymer (i.e., PVP) on the NLO response of the nanohybrids was found to be negligible, implying that the increase of the NLO parameters is exclusively attributed to the increase of the nanoparticle’s concentration.

Reviewer 2 Report

The main question addressed by the research is the synthesis, characterization, optical limiting, and nonlinear optical properties of iridium-based nanohybrids. The study aims to investigate the effects of different synthetic protocols and compositions on the chemical composition, morphology, and optical properties of the nanohybrids. The research specifically focuses on the variations in nonlinear absorption and refractive response based on the ratio of iridium to iridium dioxide (Ir/IrO2) nanoparticles in the nanohybrid solutions.

The topic is relevant in the field as it explores the potential of iridium-based nanohybrids for photonic and optoelectronic applications. It addresses a specific gap in the field by examining the influence of different synthetic methods and compositions on the nonlinear optical properties of the nanohybrids. This information can be valuable for tailoring the properties of nanohybrids to meet specific requirements in photonics and optoelectronics.

There is a problem with the numbers given for the molar masses in lines 93-95:

Iridium(III) chloride hydrate (IrCl3·3H2O; Mw = 298.58 gmol-1
polyvinylpyrrolidone (PVP; average Mw = 1,300,000 gmol-1
Sodium hydroxide (NaOH; Mw = 40,00 gmol-1

Conventionally the decimal symbol is the period sign (.) and the thousand separator is the comma (,). This looks good the first two cases. It makes sense that the polymer has an average value, and it is very high: one million three hundred thousand.

On the other hand, for NaOH the correct value is 40.00 gmol-1, i.e. the comma should be replaced by a decimal symbol.

The references are numerous and appropriate.

My comment above considering the decimal symbol is so obvious, that the editor may decide that it is not necessary for the authors to prepare a revised version, just change this. In this case I would recommend acceptance.

Author Response

Reviewer 2

Comments and Suggestions for Authors

“The main question addressed by the research is the synthesis, characterization, optical limiting, and nonlinear optical properties of iridium-based nanohybrids. The study aims to investigate the effects of different synthetic protocols and compositions on the chemical composition, morphology, and optical properties of the nanohybrids. The research specifically focuses on the variations in nonlinear absorption and refractive response based on the ratio of iridium to iridium dioxide (Ir/IrO2) nanoparticles in the nanohybrid solutions.

The topic is relevant in the field as it explores the potential of iridium-based nanohybrids for photonic and optoelectronic applications. It addresses a specific gap in the field by examining the influence of different synthetic methods and compositions on the nonlinear optical properties of the nanohybrids. This information can be valuable for tailoring the properties of nanohybrids to meet specific requirements in photonics and optoelectronics.”

Comment 1: “There is a problem with the numbers given for the molar masses in lines 93-95:

Iridium(III) chloride hydrate (IrCl3·3H2O; Mw = 298.58 gmol-1 polyvinylpyrrolidone (PVP; average Mw =1,300,000 gmol-1 Sodium hydroxide (NaOH; Mw =40,00 gmol-1

Conventionally the decimal symbol is the period sign (.) and the thousand separator is the comma (,). This looks good the first two cases. It makes sense that the polymer has an average value, and it is very high: one million three hundred thousand.

On the other hand, for NaOH the correct value is 40.00 gmol-1, i.e. the comma should be replaced by a decimal symbol.”

Authors’ Reply: We thank the reviewer for pointing this out. Following reviewer’s suggestion, the molar mass in the case of NaOH was corrected, by substituting comma with the period sign.

Comment 2: “The references are numerous and appropriate.

My comment above considering the decimal symbol is so obvious, that the editor may decide that it is not necessary for the authors to prepare a revised version, just change this. In this case I would recommend acceptance.”

Authors’ Reply: We would like to thank the reviewer for the comments about our work.

Reviewer 3 Report

This article synthesized three iridium (Ir)-based nanohybrids with different compositions, and studied their nonlinear optical (NLO) response and optical limiting (OL) efficiency, which provided some valuable information. However, there are some issues needed to be improved.

1. Figure 2 shows TEM images from different synthesis methods, the author said that ‘For sample I (Ir/IrO2), the nanoparticles could be seen only clustered into larger agglomerates’, but the presentation in the TEM is not clear enough.

2. In the introduction section, the author mentioned that “In that context, for the first time in this study”. The author should double check whether it is for the first time.

3. Figure 3 shows the proportion of each element in different samples analysed by EDX spectroscopy. However, three samples exhibit different image range sizes and the content of the element Ir. For this phenomenon, the author should provide the corresponding explanations.

4. In Table 1 and Table 2, some data is obtained from calculation, but not from the experiments. Whether the author can provide more experiment to proof the correctly of these data.

Minor editing of English language required

Author Response

Reviewer 3

Comments and Suggestions for Authors

“This article synthesized three iridium (Ir)-based nanohybrids with different compositions, and studied their nonlinear optical (NLO) response and optical limiting (OL) efficiency, which provided some valuable information. However, there are some issues needed to be improved.”

Comment 1: “Figure 2 shows TEM images from different synthesis methods, the author said that ‘For sample I (Ir/IrO2), the nanoparticles could be seen only clustered into larger agglomerates’, but the presentation in the TEM is not clear enough.”

Authors’ Reply: We thank the reviewer for this useful comment, which allows to better explain our work. From most of these samples TEM images were very difficult to obtain, due to the extremely small nanoparticle sizes and the destruction of the polymer layer by the electron beam. From Fig. 2, we selected a very small agglomerate for sample I, in order to be able to provide a clearer image of individual nanoparticles. An additional image of a larger agglomerate of the same sample has been included in the Supporting Information of the revised manuscript (Figure S2).

Moreover, the above sentence has been modified as follows:

“For sample I (Ir/IrO2), the nanoparticles could be seen only clustered into larger agglomerates overlapping each other (see also Figure S2 in Supporting Information), unlike sample III (PVP-Ir/IrO2), where at least parts of the sample were found in almost perfect monolayers.”

Comment 2: “In the introduction section, the author mentioned that “In that context, for the first time in this study”. The author should double check whether it is for the first time.”

Authors’ Reply: In response to reviewer’s comment, the phrase “for the first time in this study” was deleted.

Comment 3: “Figure 3 shows the proportion of each element in different samples analysed by EDX spectroscopy. However, three samples exhibit different image range sizes and the content of the element Ir. For this phenomenon, the author should provide the corresponding explanations.”

Authors’ Reply: We thank the reviewer for this comment. EDX is not per se dependent on the size of the area measured. EDX is also a semiquantitative method, that is why ratios between elements should be accurate enough, while absolute values (to which the area size would certainly contribute an influence) are not. In these cases, the contributions to Ir stem only from the nanoparticles, while the contributions for C come both from PVP (for samples II and III) as well as (to a very large amount) from the carbon grid, as the penetration depth of EDX is far higher than, for example, for XPS. Since the contribution to C from PVP is almost neglectable in this case, that means that a larger percentage of Ir reflects a larger number of nanoparticles in the 3-dimensional space measured. For sample III since there is more or less a monolayer of nanoparticles, and not a very dense one (in xy-direction), the smaller percentage of Ir can be easily justified. Sample I also contained stacked layers of particles, which makes the relative number of particles in this area much higher, and thus the Ir content for this sample is higher. For sample II the particles cannot be seen very well due to the large amount of PVP coating, nevertheless it is possible to draw the same conclusions, as the ones mentioned above, i.e. since the relative amount of Ir is higher, the nanoparticles most probably form stacked layers.

Based on the above, additional explanation regarding EDX analysis was included in the revised text. Moreover, Figure S2 was added in Supporting Information.

Comment 4: “In Table 1 and Table 2, some data is obtained from calculation, but not from the experiments. Whether the author can provide more experiment to proof the correctly of these data.”

Authors’ Reply: We thank the reviewer for this comment. All the data listed in Tables 1 and 2 are reported in the related literature, which is also provided in these Tables, and were experimentally determined in the cited works. So, we have not performed any further investigation or experiments. We adopted the values provided by these studies.  

Comment 5: “Comments on the Quality of English Language

Minor editing of English language required”

Authors’ Reply: Following the Reviewer’s comment, we went through the manuscript carefully and corrected some grammatical errors we found.

Round 2

Reviewer 1 Report

I have no further comments. 

Reviewer 3 Report

The author has solved all my questions. I suggest to accept this manuscript in the current form.